# Interaction between Neurons and the Oligodendroglial Lineage in Multiple Sclerosis and Its Preclinical Models

**DOI:** 10.3390/life11030231

**Published:** 2021-03-11

**Authors:** Vasiliki Pantazou, Thomas Roux, Vanessa Oliveira Moreira, Catherine Lubetzki, Anne Desmazières

**Affiliations:** 1Paris Brain Institute (ICM), Sorbonne Université, CNRS, Inserm, GH Pitié-Salpêtrière, 47 boulevard de l’Hôpital, 75013 Paris, France; vasiliki.pantazou@icm-institute.org (V.P.); thomas.roux@aphp.fr (T.R.); v.oliveira-moreira@icm-institute.org (V.O.M.); catherine.lubetzki@aphp.fr (C.L.); 2Service de Neurologie, Centre Hospitalier Universitaire Vaudois, 46 Rue du Bugnon, 1011 Lausanne, Switzerland; 3Assistance Publique-Hôpitaux de Paris, Neurology Department, Pitié Salpêtrière University Hospital, 75013 Paris, France

**Keywords:** oligodendrocytes, glia, myelin, neuron, myelination, multiple sclerosis

## Abstract

Multiple sclerosis (MS) is a complex central nervous system inflammatory disease leading to demyelination and associated functional deficits. Though endogenous remyelination exists, it is only partial and, with time, patients can enter a progressive phase of the disease, with neurodegeneration as a hallmark. Though major therapeutic advances have been made, with immunotherapies reducing relapse rate during the inflammatory phase of MS, there is presently no therapy available which significantly impacts disease progression. Remyelination has been shown to favor neuroprotection, and it is thus of major importance to better understand remyelination mechanisms in order to promote them and hence preserve neurons. A crucial point is how this process is regulated through the neuronal crosstalk with the oligodendroglial lineage. In this review, we present the current knowledge on neuron interaction with the oligodendroglial lineage, in physiological context as well as in MS and its experimental models. We further discuss the therapeutic possibilities resulting from this research field, which might allow to support remyelination and neuroprotection and thus limit MS progression.

## 1. Introduction

Along with species evolution, there is an increasing complexity in the central nervous system (CNS) organization with, in particular, the acquisition of myelination [1,2], allowing for the efficient conduction of the nerve influx along myelinated fibers [3]. The choice of the axon to be myelinated, the thickness and length of the myelinated internode—even the pattern of myelin deposition along the axon—are not randomly selected but are a rather complex, carefully orchestrated collaboration of multiple cell types, with the two main actors in the CNS being the neuron and its myelinating cell—the oligodendrocyte (OL).

OLs enwrap axons with a multilayered lipid-rich membrane, the myelin. Myelin ensures electrical insulation allowing fast saltatory conduction along the axon [4], but also provides trophic and energetic support through delivery of lactate and pyruvate by oligodendrocytes [5,6]. In vitro and in vivo studies have suggested that a single mature OL can produce 5000 µm^2^ of myelin per day [7] and myelinate about 15 internodes in zebrafish [8] and up to 60 in rodents [9,10]. Establishing stable myelin sheaths can take from 2–3 h in the zebrafish [8] to a few days in rodents [11].

Although myelination can occur in vitro in the absence of axons—even around inert fibers such as electron-spun nanofibers [6,12]—neurons modulate myelination in vivo, where selective myelination has been observed among axons [13]. A typical example is the presence of the myelinated descending motor neurons in the medial forebrain in close vicinity to the not myelinated nigrostriatal dopaminergic neurons [14]. Axonal diameter is a key determinant to whether or not an axon will be myelinated with axons over 0.4 μm being preferentially myelinated in the CNS [15,16,17]. Numerous studies show a role of neuronal activity, growth factors and various signaling pathways in myelination regulation [18,19,20,21,22,23,24,25], although many aspects of the relationship between neurons and the oligodendroglial lineage are not fully elucidated yet.

This constant dialog between the CNS myelinating lineage and neurons is not only crucial for the development of the nervous system but is maintained throughout life and disease. A clear example of where this dialogue can be impaired is multiple sclerosis (MS), the most frequent demyelinating, inflammatory and neurodegenerative disease of the human central nervous system (CNS). MS affects more than two million people worldwide and is the leading cause of clinical disability and long-term neurological handicap amongst young adults [26,27]. In MS, myelin and/or the OL lineage are the target of an immune-mediated injury characterized by attacks disseminated throughout the CNS spatially and temporally [28]. As a result of these attacks, focal demyelination and neuronal damage are observed early in the disease course, but various degrees of remyelination do occur and allow for improved functional outcome [29,30,31,32,33,34].

In the present review, we summarize the current knowledge on neuron–OL interaction, in both demyelinating and remyelinating context in MS and its experimental models, concluding on the therapeutic possibilities resulting from this research field.

## 2. The Organization and Functioning of the Myelinated Axon

The myelinated axon is organized in internodes—axonal segments electrically insulated by myelin—alternating with small unmyelinated excitable domains enriched in voltage-gated sodium and potassium channels, called the nodes of Ranvier [35,36]. The ratio of the axonal diameter to the diameter of the myelin sheath added to the axon is called g-ratio and is used as an index of axonal myelination. Myelin sheath decreases membrane capacitance and increases membrane resistance. By doing so, it stabilizes the ionic charge, as the axolemma is depolarized during action potential regeneration and propagation at the nodes [4,21,37,38]. This leads to the so called saltatory conduction along myelinated axons, a function acquired along the evolution of species, allowing for higher velocities [1,39].

Recent studies suggest that besides this rather passive role on signal conduction, OLs actively participate in shaping electrical transmission [40,41]. Axonal proteins also regulate myelin deposition along the axon. Examples of such modulation include inhibition of myelin ensheathment by PSA-NCAM [42]. Furthermore, genetic deletion of some molecules implicated in paranodal and internodal axo-glial junctions, such as Cadm4 and Caspr, lead to the formation of axons with multiple layers of myelin sheaths, loose paranodal junctions or internodes of altered length [43,44].

Besides rapid conduction at the single neuron level, differences in myelin segment length and thickness may allow to create a synchronized network activity translating to higher cognitive functions [18,45,46,47,48,49,50]. Evolution of myelin segment characteristics and pattern have further been observed within hours following motor training or social isolation indicating that myelin—much like neurons—is a plastic structure able to adapt brain function to environmental stimuli [45,51,52,53].

## 3. Neuronal Activity as a Mediator of Myelin Plasticity

Even prior to becoming myelinating cells, OPCs sense their environment by monitoring electrical activity of surrounding neurons through NMDA, AMPA/kainate and GABA receptors [20,54,55,56,57,58]. Accumulating evidence suggests that OPCs maintain the capacity to communicate with neighboring neurons through non-synaptic junctions during different stages of development and maturation to myelinating oligodendrocytes [24,56,58] (for review see [59]). Examples of such a communication are the expression of NMDA receptors in mature OL, binding axon-released glutamate in the optic nerve of the rat [23], or the Ca^2+^ signals modulating myelin sheath length during myelination in zebrafish larvae [60].

Why is it important for the oligodendroglial lineage to maintain the capacity to monitor neuronal activity? Early in vitro experiments demonstrated that altering the sensing of neuronal activity interfered with myelination by using neurotoxins such as tetrodotoxin or α−scorpio toxin, which specifically either block or increase the probability of opening of voltage-gated sodium channels, respectively, Demerens et al. demonstrated that myelination was inhibited in electrically silent axons (tetrodotoxin) or enhanced by increasing neuronal activity (α−scorpio toxin) [19]. Later in vitro experiments in dorsal root ganglion neurons by Wake et al. showed that OPCs can detect axonal spiking and glutamate vesicular release and differentiate into myelinating OLs [56]. Further proof that neuronal activity stimulates OPC differentiation came more recently with the use of optogenetics in transgenic mice with myelination occurring preferentially on electrically active axons [21,61]. The development of two photon microscopy has allowed for in vivo live imaging, further expanding our knowledge. In zebrafish larvae, Mensch et al. have in particular shown that—similarly to in vitro experiments—individual OLs are generally capable to detect electrical activity through vesicular release by their neighboring neurons and, as a response, modulate their myelinating capacity [25].

## 4. Oligodendrocyte Lineage and Neuron: A Story of Mutual Dependence

Myelination has a dramatic impact on neuronal function and organization, as the mere act of myelin ensheathment around an axon improves the chances of survival of both cells. Indeed, OLs that do not manage to ensheath axons degenerate [18,62]. On the other hand, loss of OLs disrupts the normal nodal and paranodal architecture and leads to severe axonal pathology [50]. Furthermore, OLs promote neuronal survival though secretion of neurotrophic factors such as IGF-1 and GDNF [63] and provide metabolic support for the axon through delivery of lactate [64]. OLs are indeed highly enriched in monocarboxylase transporter 1 (MCT-1), the most abundant lactate transporter in CNS [17,65]. Lactate fuels axonal mitochondria, and by doing so, crucially supports highly energy-demanding axonal transport. Highlighting the necessity of this interaction for axonal integrity, MCT-1 knockout mice show age-dependent neuronal loss without evident demyelination [5]. Most recently, in an in vitro culture of mouse corpus callosum slices, Meyer et al. demonstrated that OLs fueled neuronal activity through glucose rather than lactate delivery [66], suggesting that OL–neuronal interactions are adapted to their local environment as they vary among cell types, locations and energetic needs. Interruption of this axo-glial metabolic support leads to deficits in information processing, even in the absence of demyelination, as demonstrated in the elegant experiments conducted by Moore et al. [67].

## 5. Altered Myelin and Axonal Damage: Lessons from the Animal Models

Studies in animal models have provided compelling evidence that axonal damage can result from an altered myelin. Myelin-associated glycoprotein (MAG) is one of the four major myelin proteins (along with myelin proteolipid protein—PLP, myelin basic protein—MBP and 2′,3′-cyclic-nucleotide phosphodiesterase—CNPase). MAG is expressed in the inner layer of myelin sheath and binds to axonal gangliosides GD1a and GT1b, establishing the axo-glial contact. In MAG-deficient mice, both motor development and myelin content appear normal. In closer inspection, subtle changes are observed, such as in neurofilament phosphorylation and spacing, and these changes ultimately lead to axonal degeneration in old age [68].

In PLP-deficient mice, myelin is still produced and packed thickly but is highly unstable. As a result, these mice present with normal early development but after 6–8 weeks they show focal axonal swellings containing membranous organelles, multivesicular bodies and mitochondrial dysfunction. By 18 months of age, they have significant motor impairment and extensive axonal degeneration [69]. Similar findings are reported in some patients with PLP mutations (Pelizaeus Merzbacher disease), who show CNS white matter disease, with axonal degeneration as measured by proton magnetic resonance spectroscopy analysis of N-acetyl-aspartate (NAA) [70].

On the other side of the spectrum, mutations of MBP result in unmyelinated axons or thin non-compacted myelin sheaths. Life span is reduced in MBP mutant rodents (up to 4 months for the Shiverer mice and up to 9 months for the Evans Shaker rats) [71,72]; nonetheless, even in terminal stages, demyelination is not followed by axonal damage. In these models, upregulation of neurotrophic factor expression in the remaining OLs and astrocytes is found [73], suggesting that degeneration may be prevented by compensatory mechanisms through axo-glial communication.

## 6. Myelin Insult and Neuronal Damage in Multiple Sclerosis

Focal demyelination of the CNS white matter is the hallmark of MS, as described by Charcot over 150 years ago [74,75]. Regardless of the perspective from which it is viewed (disease pathology, clinical and radiological findings, histopathological findings, outcomes), MS remains a highly complex disease. Clinically, the majority of patients will initially present with a single, self-limited, neurological episode, described as a clinically isolated syndrome, that typically involves the optic nerve, brainstem, or spinal cord. MS diagnosis is possible even at this early stage in the presence of typical findings on an abnormal MRI scan and/or positive oligoclonal bands in the cerebrospinal fluid [76].

In MS, 85% of the patients present with a relapsing remitting form of the disease [27]. Active demyelination is considered the histopathological equivalent of the clinical relapse [77]. Four patterns of active lesions have been described (pattern I–IV) [78], all sharing demyelination as a common denominator with various degrees of inflammatory cell infiltration (peripheral macrophages, activated microglia, T lymphocytes, complement, plasma cells) and OL loss. OL loss or myelin destruction will lead to a conduction block, which can translate to a neurological deficit, and if prolongated, will also lead to neuronal damage [79] (Figure 1). Depending on the number of adjacent internodes being demyelinated, nerve conduction velocity can be decreased or completely blocked. So far and to the best of our knowledge, how many internodes need to be demyelinated to cause a conduction block still remains to be established.

However, axonal loss and demyelination in absence of macrophage infiltration can be found in about 15% of patients who develop progressive onset MS described as primary progressive MS (PPMS) [27,77]. Furthermore, focal changes have been documented in normal-appearing white matter of MS months to years before the appearance of MRI gadolinium-enhanced lesions. Altered axonal organization with changes in the distribution of sodium and potassium channels, early alterations of paranodal structures [35,80,81,82,83], focal axonal swelling or degeneration can be found prior to OL loss [84]. Such findings opened the discussion on whether axonal degeneration is a direct consequence of demyelination, an independent mechanism or even the prime event leading to demyelination.

## 7. Endogenous Remyelination Occurs in MS and Its Animal Models

Alongside with demyelinated axons, lesions with evidence of remyelination, also known as “shadow plaques”, are characterized by the presence of thinner myelin sheaths and shorter internodes. Post-mortem human MS tissue studies have shown that 20% of the patients showed fully remyelinated lesions [33]. For a brief history on discovery of remyelination and discussion on whether shadow plaques are signs of remyelination or, on the contrary, of ongoing demyelination, see [31,85]. Shorter internodal segments, absence of immune cell infiltration and only few microglia/macrophages within the lesion help to distinguish shadow plaques from early demyelinating lesions.

Although often extensive in animal models of demyelination, the degree to which remyelination occurs in MS patients is variable. It differs among locations, with a higher degree of remyelination in cortical and subcortical lesions compared to periventricular or cerebellar lesion [86], and among patients, as shown in histopathological studies and PET-MRI studies using myelin markers ([11C]PIB) [32,33,87,88]. Subgroups of MS patients with either extensive or limited remyelination with variable myelin content are identified, suggesting that inter-individual myelination capacity may further influence the ability of lesions to remyelinate.

Whether remyelination occurs from surviving OLs or newly differentiated OLs derived from OPCs have been studied over the last two decades and is comprehensively discussed in a recent review by [89]. The notion that remyelination requires OPC recruitment and differentiation comes from multiple observations: 1. After OL and OPC depletion by ethidium bromide induced demyelination in rodents, OPCs appear in the lesion site prior to mature OLs [90]; 2. Transitioning phenotypes of OPCs have been observed in demyelinating lesions in adult rat brain after toxic-induced demyelination. Different surface or nuclear markers can help differentiate OL lineage cells, as the proteoglycan NG2, the platelet derived growth factor receptor α (PDGFRα), OLIG1, OLIG2 and Nkx2.2 are expressed early in OPC development, while O4 and glycolipid galactocerebrosidase (GalC) appear later in immature or pre-myelinating OL and so on [91,92]; 3. After single cell OL laser ablation, an homeostatic replacement takes place by a newly differentiated OPC who produces myelin with thinner and shorter internodal sheaths, as observed by live two photon imaging, but sufficient to restore function [93]; 4. Newly differentiated OLs in a capsaicin-induced model of demyelination in zebrafish were more efficient in myelin sheath formation than OL survivors [94]; 5. In post-mortem analysis of MS brains, there was an increased density of OPCs in the subventricular zones compared to controls, which would further increase in active lesions compared to normal appearing white matter (NAWM) [95]. Furthermore, in the cuprizone model, experimental conditions favoring differentiation of subventricular zone neural stem cells into oligodendrocytes resulted in remyelination with normal thickness myelin sheath restoring normal nerve conduction [96,97].

The role of mature OLs in remyelination has been debated, since these cells do not seem to have migrating nor proliferating capacities. Nonetheless, almost 30 years ago, this notion was challenged when mature OLs grafted into newborn shiverer mouse brain produced new myelin sheaths around denuded axons [98]. The use of multiphoton imaging to track OL lineage response following cuprizone-induced demyelination in mice cerebral cortex [11] and capsaicin-induced demyelination in zebrafish [94] have provided more insight into myelin regeneration. In both studies, newly differentiated OLs produced large number of myelin sheaths forming internodes along the axon. Surviving OLs could only occasionally produce new myelin sheaths, often misplacing them to neuronal soma [94]. Although this process was less efficient, it occurred over protracted periods of time and was enhanced by training [11]. These observations in the animal models suggest that even though both mature and newly differentiated OLs can produce myelin sheaths, it is newly differentiated OLs, rather than mature OLs, that orchestrate remyelination in adult brain following injury in these models.

In humans, given the low level of oligodendrocyte generation in adult CNS tissue [99,100], it is unlikely that newly generated OLs alone can respond to the demand in myelin production. Recent works have indeed shown a very low rate of new OL generation in shadow plaques of MS patients [100], further favoring a role of pre-existing OLs for remyelination in MS. However, remyelination by mature OLs happens only at low levels in mice [11,94], underlying a potential difference in remyelination mechanism in MS and its animal models.

## 8. Remyelination Failure in MS

Even though remyelination is possible in MS, the majority of lesions show only partial remyelination or no remyelination at all [33], despite sufficient number of OL-lineage cells were present at the lesion sites [101], suggesting that other mechanisms than paucity of myelinating cells participate in successful remyelination. Repetitive rounds of demyelination-remyelination is a characteristic feature of the disease in humans [102]. Using lesion segmentation from MTR images, MRI studies have demonstrated recurrent rounds of inflammatory demyelination in pre-existing lesions, associated with lower capacities of recovery [79,88]. Furthermore, ageing has been associated with reduced ability to remyelinate [103,104,105]. Although mechanisms that lead to remyelination failure are not yet fully elucidated, what has been made obvious is that remyelination failure correlates with increased clinical disability and extensive axonal loss (Figure 1) [30,103]. This paradigm is seen in 15–30% of RRMS patients, who, over time, develop progressive disability described as secondary-progressive MS [27]. These patients show extensive axonal degeneration and little evidence of remyelination, although a subset of them may still exhibit inflammatory activity as shown clinically by relapses, and by areas with sharp limits of demyelination and a rim of macrophages at the lesion edge in pathology [77], that mediate active ongoing myelin breakdown and axonal damage [88]. The rim is also identifiable in MRI studies as a persistent phase rim [106] or an iron rim at the lesion edge [107], which can provide a useful prognostic tool in MS patients [108].

Furthermore, in remyelinated lesions, axonal density is higher with a less extensive degree of axonal damage than in non-remyelinated lesions (as measured by β-amyloid-precursor protein staining), suggesting a positive effect of new myelin sheath formation to axonal preservation [109]. Similar findings are reported in experimental autoimmune encephalomyelitis (EAE), an inflammatory animal model of MS, in which remyelination confers preservation of axonal density and improvement of functional recovery and outcome [110].

It is thus essential to better understand how to promote an adequate neuro-glial crosstalk to support remyelination and limit the progression of the disease.

## 9. After a Demyelinating Insult, What Are the Required Steps for New Myelin Sheath Formation?

In order to promote repair in MS, it would be of interest to increase OPCs recruitment to the lesions and favor their differentiation in myelinating OLs, as well as promote remyelination by surviving OLs. Indeed, the prevailing opinion is that efficient myelin repair requires activation and recruitment of OPCs to demyelinated areas, followed by their differentiation into mature OLs and target recognition of the axon to remyelinate (Figure 2). At the lesion site, newly differentiated OLs along with surviving OLs proceed to new myelin sheath formation and production of stable myelin sheaths [30,89]. As discussed below, each of these steps requires the interaction between neurons and the oligodendroglial lineage.

Following activation, OPCs migrate to the lesion site attracted by guidance cues [47,111]. Growth factors like insulin-like growth factor 1 (IGF-1) and glial cell line-derived neurotrophic factor (GDNF) have also been shown to act in OPC recruitment [47,89]. Among attractant cues, Semaphorin 3F has been identified both in in vitro and in vivo models. In contrast, Netrin was shown to act as a repellent for OPCs [112], as well as Semaphorin 3A [111]. Neurons further secrete adenosine and stimulate neighbor astrocytes to produce leukemia inhibitory factor or CNTF to promote OPC differentiation into myelinating OLs [113,114]. The next step to remyelination is careful selection of the axon to remyelinate [18,62]. As previously stated, neuronal activity also plays a role in remyelination [20,61]. OPCs are not only sensitive to the presence of neuronal activity, but also to the pattern of activity, which modulates differently their proliferation and differentiation [55]. Furthermore, following a demyelinating insult, dispersion of voltage-gated sodium channels is observed along the axon [80,81,115]. In a very recent study, it has been shown, using a cuprizone-induced demyelination model, that βIV spectrin, which forms a complex with Ankyrin-G to link voltage-gated channels to actin cytoskeleton at the nodes of Ranvier, remains however clustered after demyelination in the cortical layers of the somatosensory cortex of adult mice, which could provide reclustering cues for nodal voltage-gated ionic channels [116]. Interestingly, multiple studies conducted by us and others reported nodal reclustering can occur prior to (re)myelination and suggest they could impact conduction velocity and myelin initiation guidance [35,117,118,119].

In contrast, pathways inhibiting OPC maturation and recruitment have also been identified such as axonal expressed Jagged-1. Jagged-1 protein binds OPC expressed NOTCH-1, inhibiting their differentiation [119]. More recently, the Wnt pathway and the mechano-responsive ion channel Piezo1 activation have also been identified as OPC differentiation inhibitors, possibly involved in age-related remyelination capacity decline [30,89].

Moreover, the perineuronal extracellular matrix (ECM) could also modulate remyelination abilities in MS. Indeed, hyaluronan, as well as chondroitin sulfate proteoglycans (CSPGs) are described to accumulate at the border of actively demyelinating plaques, where they inhibit the migration and differentiation of oligodendrocyte precursor cells (OPCs) and remyelination [120]. Enhanced OPC differentiation and remyelination were further observed in cuprizone and EAE mice models when suppressing CSPGs activity [121,122].

Better understanding the complex dialogue between neurons and the oligodendroglial lineage and how it is balanced in the context of their direct environment will thus be required to adequately promote axon remyelination in MS.

## 10. Other Cellular Contributors to Successful Remyelination

It is now clear from the abundant literature on the subject that the process of myelin repair requires a carefully orchestrated collaboration between multiple players (Figure 2).

Whether astrocytes bring more damage or inhibit repair is still under debate [123], although evidence from studies using ethidium bromide-induced demyelination speaks favorably for their role during remyelination, as elimination of astrocytes resulted in absence of CNS remyelination [124]. In addition, phagocytosis of myelin debris by surrounding microglia/macrophages promotes remyelination [125,126,127]. Their role further extends beyond phagocytosis, as microglial switch to a pro-regenerative profile promotes remyelination, and maintenance of a more pro-inflammatory phenotype is deleterious for repair. Microglia secrete various factors that can modulate the oligodendroglial lineage and promote (re)myelination [128,129,130,131] (Figure 2). The CXCL12 chemokine is expressed by microglia in MS lesions and has been described to modulate OPCs chemoattraction and differentiation [132]. Microglia is also described as a major source of iron for OPCs during developmental myelination, promoting their proliferation and differentiation and could further play a role in remyelination [133]. Furthermore, both astrocytes and microglia can perceive neuronal activity, suggesting they could participate in an indirect dialogue between neurons and the oligodendroglial lineage (for review, see [59]).

It has further been shown that failure in OPCs recruitment and differentiation increases with age [105,134,135], and the Franklin laboratory showed this could be reverted using a heterochronic parabiosis experiment, in which old mice shared systemic circulation with younger mice. After LPC induction of spinal cord demyelination, they observed a higher rate of remyelination in these mice comparing to the isochronic controls (old mice sharing systemic circulation with their age homologues), due to the recruitment of “younger” blood-derived monocytes from their partners. Moreover, the importance of monocyte/macrophage recruitment in remyelination was highlighted using mice deficient in CCR2, a major player in macrophage recruitment, in which a significant decrease of proliferating OPC was observed when compared to controls [104].

## 11. Myelin Repair: From Animal Models to Human Translation

### 11.1. The Challenges of Myelin Repair Strategies.

The past two decades have been an era of exponential growth of disease-modifying immunotherapies acting on the inflammatory component of the disease, preventing these recurrent episodes of inflammation [27]. As a result, patients experience fewer or no relapses, show less or no gadolinium-enhanced lesions and less disease progression (described as NEDA-3 status) [136]. The recent epidemiological studies have shown that the rate of conversion to secondary progressive MS has been reduced from 50% to less than 30% over the last 10 years [27].

However, these therapies are still insufficient to block this progressive phase, and half of the patients will suffer a disability accumulation [27,30,137,138]. In the light of this unmet need, the scientific community is now focusing its efforts on ways to promote remyelination in an effort to ensure neuroprotection [139].

A key challenge will be that at any given time point of the disease, the patterns of demyelination and axonal degeneration are different among patients or even in different brain regions of a given patient [140]. This heterogeneity in MS brain with variable—but without a doubt concomitant—presence of OL, OPC and axonal dysfunction from early on, defines the therapeutic challenge of the disease. Regarding this aspect, a better understanding of the basic mechanisms underlying MS pathology and repair is required.

In parallel, in the quest of remyelination candidate therapies, an important step is coming up with the development of screening strategies that are both adapted and available. In vitro assays of rodent-derived OPCs or micropillar arrays [141] provide a fast way to test the promyelinating effect of bioactive molecules but they offer little insight in the molecules’ physiological mechanism of action. Animal models including toxin or chemically-induced demyelination followed by spontaneous remyelination bypass the aforementioned problem. Among them, zebrafish larvae and *Xenopus laevis*, allow for a robust large screening platform, while EAE models have the advantage of resembling the closest to the human MS disease but are not suitable for the screening of large numbers of compounds (for review, see [30]).

Using these different approaches, a multitude of compounds has been tested following two strategies: revisiting existing drugs for their potential role on myelination or developing new molecule-targeting pathways of OPC differentiation and axo-glial interaction.

### 11.2. Repurposing Existing Drugs for Their Remyelinating Potential

A great example of a molecule repurposed for its promyelinating potential is clemastine fumarate. Clemastine is a first-generation histamine H1 receptor antagonist used since the early 1990s in the treatment of allergies. Clemastine crosses the blood–brain barrier and was found to promote OPC differentiation and myelination in a high-throughput in vitro screening and its efficiency in remyelination was confirmed in mouse in vivo following LPC-induced focal demyelination [141]. The molecule was then tested in the cuprizone-induced model of demyelination, with similar results [142]. Its effect on OPC maturation is thought to be mediated by an antimuscarinic action in the M1 muscarinic acetylcholine receptor Chrm1 [110]. From clinical biology to human pathology, clemastine was tested in chronic optic neuritis patients in a phase II trial [143] (ReBUILD trial) where it met its primary outcome and showed a small reduction in evoked potential latency but failed to improve clinical outcome. The drug is now tested in acute neuritis patients in the ReCOVER trial (NCT02521311).

Other than clemastine, a large number of molecules have been tested in preclinical studies for their potential benefit in myelination. Less than 20 have demonstrated significantly enhanced remyelination in animal models, either by promoting OPC differentiation or by reducing microglial and astrocyte activation [139]. Such drugs involve anticholinergics and antimuscarinics (GSK239512 [144], benzatropine [145]), antiemetics (domperidone [146], antipsychotic drugs (quetiapine [147]), selective estrogen modulators (progesterone, tamoxifen [148]), hormones (ACTH [149]), antifungal drugs (miconazole [150]), antidiabetic drugs (metformin [151]), immunomodulatory drugs (glatiramer acetate, siponimod, teriflunomide [151,152,153] and statins [154]. Simvastatin, a widely used statin, is one of the few of these molecules being tested in clinical trials (Table 1). In a phase II study including 70 SPMS patients (MS-STAT study), the authors report a 43% reduction of annual brain atrophy rate and a 63% reduction in the number of black holes with simvastatin treatment [155]. Less promising results were obtained in the phase IV SIMCOMBIN trial in RRMS patients, as an add-on therapy to interferon-beta, where simvastatin failed to further reduce annualized relapse rate [156] underlying a pro-regenerative rather than anti-inflammatory role of the drug.

On the other hand, not all molecules tested had a null or positive impact on myelination. Some seemed to hasten remyelination after a demyelinating insult, such as valproic acid, a first generation antiepileptic drug [157].

### 11.3. Strategies Promoting OPC Differentiation

The second approach involves molecules that have been designed to target a specific pathway in OPC differentiation or maturation. Multiple monoclonal antibodies have been developed the last two decades and are being tested in clinical studies. Studies showed that Semaphorin 4D is upregulated in OPCs upon spinal cord injury and inhibits remyelination in EAE [158,159], while using an anti-Semaphorin 4D rescues oligodendroglial differentiation, with a strong reduction of inflammatory activity in EAE. A humanized monoclonal anti-Semaphorin 4D antibody (VX15/2503) was tested in a phase I study with no major side effects [160].

Similar “from bench to clinical translation” examples include temelimab/GNbAC1, a humanized antibody directed against the envelope protein (ENV) of the human endogenous retrovirus (HERV) [161] and opicinumab, an anti-LINGO-1 monoclonal antibody [162]. Following observations in in vitro OPC cultures that ENV protein inhibited OPC differentiation via activation of Toll-like receptor 4 [163], temelimab testing completed successfully a phase I trial in 21 individuals. As a consequence, the phase IIa CHANGE-MS trial [161] and phase IIb ANGEL-MS trial (Hartung et al., presented in ECTRIMS 2019) investigated efficacy of variable regimens of temelimab on over 200 MS patients. The authors report a reduction in cortical and thalamic atrophy and a benefit in magnetization transfer ratio (MTR) in both NAWM and cerebral cortical bands, suggesting an effect on remyelination. However, although loss of LINGO-1 enhanced myelin sheath formation and myelination in preclinical studies [162] and showed a small efficacy in reducing evoked potential latency in patients with acute optic neuritis (RENEW trial) [164], two follow up phase II studies (SYNERGY, AFFINITY) failed to reach their primary endpoints (Biogen Press Communication in October 2020) [165].

Inhibition of Bruton’s tyrosine kinase (BTK), a member of the Tec family of kinases, expressed by cells of hemopoietic origin and microglia has moreover been shown to favor remyelination [166] and a phase II clinical trial in MS using a BTK inhibitor has shown beneficial outcome [167]. As a result, four phase III trials (GEMINI I & II, PERSEUS, HERCULES) are ongoing examining BTKs’ efficacy in different MS populations (relapsing, primary and secondary progressive).

More recently, new approaches directed against the immune component in relation to its impact on OL lineage have emerged. Among them bexarotene/IRX4204, a specific agonist of retinoic acid receptor gamma (RXR-γ), reduced EAE severity by decreasing Th17 responses and boosting OPC differentiation [168]. The results of the phase IIa study including 52 active MS patients randomized to receive either bexarotene or placebo were recently communicated (ECTRIMS, September 2020). The investigators reported an advantage of bexarotene over placebo in reducing mean magnetization transfer ratio in a subset of MRI lesions, although drug-treated patients experienced serious side effects.

Furthermore, the use of anacardic acid improved the clinical scores of both cuprizone-induced demyelination and EAE through induction of IL-33 and upregulation of genes involved in myelin protein synthesis [169]. Melero-Jerez et al. very recently showed in the EAE model that myeloid-derived suppressor cells directly influence OPC survival, proliferation, and differentiation through upregulation of osteopontin [170].

These data highlight a crucial pathogenic interaction between innate immunity and the CNS myelination potential, opening new ways to promote effective myelin preservation and repair in MS patients. Along with the results of the ongoing studies, future directions include translation to clinical trials for the molecules showing remyelinating potency in the animal models of demyelination, as well as potential for cell grafts promoting remyelination, including fascinating approaches such as administration of encapsuled OPCs within hydrogel particles via the nose-to-brain pathway [171].

## 12. Conclusions

Even if a solid conclusion from our current knowledge on remyelination strategies is premature, the results of this research field are exciting for therapeutic development in multiple sclerosis. Understanding the complex interaction of neurons with the oligodendroglial lineage throughout development and disease and how it is modulated by surrounding glial cells may be the key determinant to successfully promote remyelination, ensure neuroprotection and reduce disability progression, which is the main unmet need in multiple sclerosis.

## Figures and Tables

**Figure 1 life-11-00231-f001:**
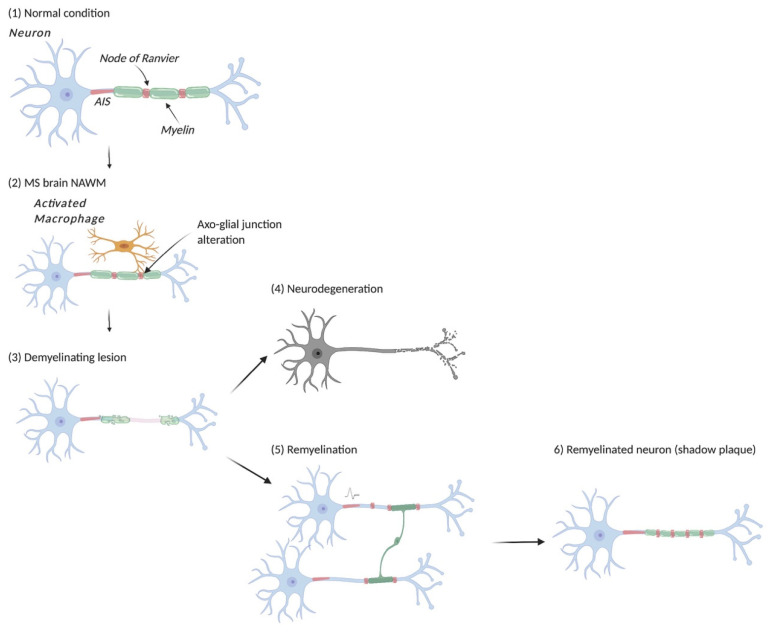
Schematic representation of myelin and axonal domain alteration and reformation in MS and its experimental models. Compared to normal condition (**1**), myelinated axon first undergo alteration at the nodal area in NAWM (normal appearing white matter) (**2**), with macrophage attack, paranodal axo-glial junction disruption and the appearance of reversible focal axonal damage (FAD). In demyelinating lesions (**3**), myelin degradation and nodal marker redistribution are associated with reduced conduction velocity or conduction blocks and associated functional deficits. These insults can lead to neurodegeneration with time (**4**), but an endogenous remyelination process exists (**5**), which is promoted by neuronal activity, and can lead to the restoration of myelinated fiber organization and efficient axonal conduction (**6**). AIS: axon initial segment.

**Figure 2 life-11-00231-f002:**
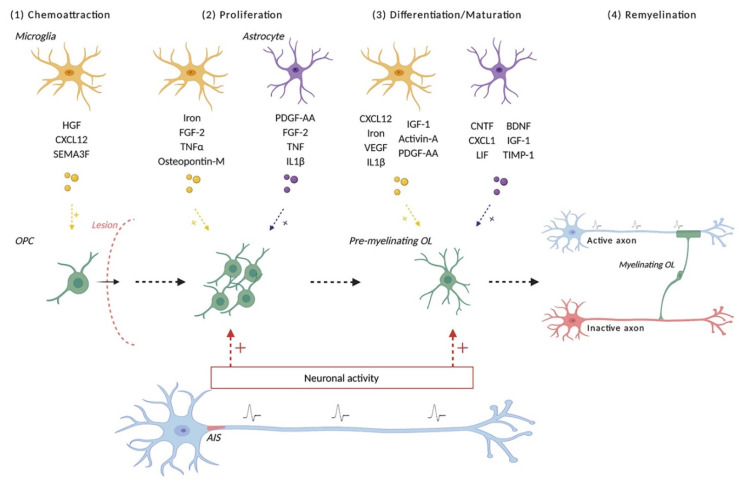
Schematic representation of modulators of the remyelination process. Remyelination depends on the recruitment of OPCs at the demyelinated lesion and their differentiation into oligodendrocytes, which will remyelinate demyelinated axons. Neuronal activity promotes multiple steps of this pathway, including the myelination process per se (choice of the axonal target, myelin stabilization). Other key players correspond to microglia and astrocytes, which secrete various factors modulating the different steps of the (re)myelination process. OPC: oligodendrocyte precursor cell; OL: oligodendrocyte.AIS: Axon Initial Segment.

**Table 1 life-11-00231-t001:** Molecules tested in clinical trials for their remyelinating potential. AAR: Annualized Atrophy Rate in MRI; ARR: Annualized Relapse Rate; ACTH: Adrenocorticotropic Hormone; BTK: Bruton’s Tyrosine Kinase; CDP: 6 month Confirmed Disability Progression; EAE: Experimental Autoimmune Encephalomyelitis; EDSS: Expanded Disability Status Scale; HERV: Human Endogenous Retrovirus; IFN: Interferon Gamma; MTR: Magnetization Transfer Ratio; MWF: Myelin Water Fraction; LPC: Lysophosphatydilcholine-Induced Demyelination; mAb: Monoclonal Antibody; NfL: Neurofilament; ODRS: Overall Disability Response Score; OL: Oligodendrocyte; OPC: Oligodendrocyte Progenitor Cell; RNFL: Retinal Nerve Fiber layer; T25FW: Timed 25 Feet Walk Test; VEP: Visual Evoked Potential. Clinical trial data listed from ClinicalTrials.gov accessed on 7 March 2021.

Drug Name	Mode of Action	Preclinical Studies		Clinical Studies			
		Model	Effect	Study	Population	Outcome	Results
**ACTH**	Corticotropin hormone	Cell culture	OPC differentiation/OL maturation	Phase IV (NCT02446886)	RMS	MWF over 12 months	Completed; results pending
**Bexarotene/** **IRX4204**	Retinoid X receptor γ agonist	EAE	OPC differentiation/reduced EAE severity	Phase IIa ISRCTN14265371	RMS	MTR; VEP latency	Improvement in MTR in a lesion subset; reduction in VEP latency ; serious side effects
**Clemastine fumarate**	H1 and M1/M3 receptors antagonist	Micropillar array screen; LPC; cuprizone	OPC differentiation/OL maturation/reduced EAE severity	Phase II ReBUILD trial Phase II Recover Trial	Chronic optic neuritis Acute optic neuritis	P100 VEP latency P100 VEP latency; RNFL thickness; EDSS; MWF	Reduced P100 VEP latency in chronic optic neuropathy without clinical improvement Ongoing
**Domperidone**	D2/D3 dopamine receptor antagonist	LPC	OL maturation	Phase II NCT02308137 Phase II NCT02493049	SPMS RRMS	T25WF MRI lesions; EDSS	Completed; results pending Completed; results pending
**GSK239512**	H3 receptor antagonist	Cuprizone	OPC differentiation/OL maturation	Phase II NCT01772199	RRMS	MTR activity	Small improvement in lesion remyelination
**Opicinumab**	mAb against LINGO 1	EAE; LPC	OPC differentiation/OL maturation	Phase IIa RENEW trial Phase IIb SYNERGY trial Phase II AFFINITY trial	Acute optic neuritis RMS RMS	P100 VEP latency; RNFL thickness; MRI lesions EDSS progression ODRS; MTR	Minor reduction in P100 VEP latency No improvement vs. placebo No improvement vs. placebo
**Quetiapine**	Antipsychotic drug	Cuprizone	OL maturation; inhibition of activated microglia	Phase I/II NCT020087631	RMS	Tolerance; EDSS	Completed; results pending
**Temelimab/GNbAC1**	mAb against HERV envelope protein	In vitro OPC cultures	OPC differentiation	Phase IIa CHANGE-MS trial Phase IIb ANGEL-MS trial	RMS RRMS	MRI lesions; morphometry	Significant reduction on cortical and thalamic atrophy; no decrease in new lesions
**SAR442168** **Evobrutinib/M2951**	BTK inhibitor	LPC demyelinized cultures	1.7× improved remyelination comparing to placebo	Phase III HERCULES trial Phase III PERSEUS trial 2 PHASE III GEMINI1 and GEMINI2 Phase II NCT02975349	SPMS PPMS RMS RMS	CDP;AAR; NfL MRI, ARR, EDSS	Ongoing Reduction in enhancing lesions and ARR, no effect on EDSS, transaminase elevation
**Simvastatin**	HMG-CoA reductase inhibitor	EAE	OPC differentiation/prevent EAE	Phase II MS-STAT trial Phase III MS-STAT2 trial Phase IV SIMCOMBIN trial	SPMS SPMS RRMS	AAR EDSS Multiparameter clinical outcomes ARR; MRI	43% reduction of brain atrophy Ongoing No additional benefit to IFN treatment

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
