# Peer review of "Interaction between Neurons and the Oligodendroglial Lineage in Multiple Sclerosis and Its Preclinical Models"

_life, 2021, doi:10.3390/life11030231_

Round 1

Reviewer 1 Report

The manuscript presents a clearly written and useful review of the MS disease, focusing on remyelination during the disease process.

As it often happens in CNS papers, the role of the extracellular matrix (ECM) is forgotten. In CNS injuries, such as traumatic injuries and injuries due to MS, chondroitin sulfate proteoglycans (CSPGs) accumulate as major ECM components inhibiting axonal regeneration and myelination. The phenomenon has been extensively studied in traumatic injuries but in addition quite many reports are found in various MS models. Useful references are for example:  Kuboyama et al., PLos ONE, 2017 and Luo et al., Nat Commun 2018.

The authors should compile a chapter on the ECM role. 

Reviewer 2 Report

This is a very interesting and comprehensive review about therapeutic advances in multiple sclerosis based on promoting oligodendroglial cell differentiation and maturation to favor remyelination.

Minor points

Paragraph 2: The organization and functioning of the myelinated axon

Lane 85: It seems that this paragraph concerns the role of axo-glial adhesion molecules in regulating myelination pattern and nodal assembly (and maybe not myelination pattern adaptation which is the topic of paragraph 3). Lane 90: Genetic deletion of these molecules is unclear: "these" seems to be related to Neurofascin and Cadm4 that should be mentioned here.

The text or legend of Figure 2 could be more detailed about iron, chemokines, and growth factors secreted by microglia. In the text corresponding to this Figure, only SEMA3F and IGF-1 (not IFG-1 as indicated in Fig?) are mentioned. Does the neuronal activity also influence microglia in addition to the oligodendrocytes?

Lane 421: “in vitro assays…offer little insight in the molecules mechanism of action”. Do you mean physiological mechanisms of action?

Lane 474: However, the humanized monoclonal anti-semaphorin 4D antibody was tested in a phase I study with no major side effects. Why However ?,  this study has stopped after phase I ?

Table I: The legend should include definitions of the abbreviations which are not in the text. This could be useful for non-clinician readers. i.e. P100 VEP, AAR…

Correct Opicinumab in the Table

Some drugs (ACTH, GSK239512, SAR442168) are not mentioned in the text, maybe a comment could be added in the legend?

Lane 38: a single mature OL can produce 5000 µm2 of myelin per day.

Lane 88: Reference Thetiot et al. 2019a: the citation should be Glia 2019 and not BioRxiv

Lane 243: OPCs

Lane 246: platelet derived growth factor receptor a (alpha)

Lanes 263, 277: OLs instead OL

Reviewer 3 Report

This review provides an overview of the interactions at play between neurons and oligodendrocytes during the demyelination process. It is well written and easy to read. I suggest few references to be added and also some clarification in studies relating to humans (MS patients) and animal models. See my comments below:

« As a result of these attacks, focal demyelination and neuronal damage are observed early in the disease course, but various degrees of remyelination do occur and allow for improved functional outcome (Franklin and ffrench-Constant, 2017; Lubetzki et al., 2020a; Neumann et al., 2020). » Instead of (or in addition to) reviews, original works evidencing such remyelination deserve to be cited here (Patrikios et al 2006 ; Albert et al 2007 ; Bodini et al 2016)

 « Genetic deletion of these molecules leads to the formation of axons with multiple layers of myelin sheaths, wide internodes and loose par- 91 anodal junctions (Elazar et al., 2019; Klingseisen et al., 2019), suggesting that adequate  myelin formation requires signaling from a functioning axon. » This sentence is not quite clear:  what do you refer to saying « these molecules »

 « Besides rapid conduction at the single neuron level, differences in myelin segment  length and thickness may allow to create a synchronized network activity translating to  higher cognitive functions (Barres and Raff, 1999; Fields, 2008; Itoh, 2015; Simons and Trajkovic, 2006) »  Please add Karimian et al., 2019 ; Bells et al., 2019, Steadman et al 2020.

 In the paragraphs considering the interaction between oligodendrocyte and neurone survival, it could be worth citing Oluich et al 2012

 « Furthermore, in the cuprizone model, experimental conditions favoring differentiation of subventricular zone neural stem cells into oligodendrocytes resulted in remyelination with normal thickness myelin sheath restoring normal nerve conduction » Please cite the work of Xing et al., 2014

 Chapter 7 “Endogenous remyelination occurs in MS” is a bit misleading since it does not focus on MS but instead it puts together data from rodent models and from MS patients, and such a mix can lead to confusion. Here are few examples:

 « Given that adult OPCs represent only 5-8% of the whole CNS cell populations, it is unlikely that they alone can respond to the demand in myelin production (Franklin et al.,2020). This assumption is questionable and Franklin’s words are a bit misused here. Indeed what Franklin wrote concerned specifically humans « This indicates that the active growth and maintenance of myelin in white matter tracts in humans is driven by mature oligodendrocytes. Given this conclusion, it is difficult to see how the low level of oligodendrocyte generation (not the small proportion of OPC) could underlie the rapid changes in white matter/ myelin volume seen in healthy humans ». In rodents, these 5-8% of OPCs are very efficient to ensure remyelination and myelin plasticity because they can divide actively when needed. Few spared OPC can indeed repopulate the whole OLG population. Since your sentence follows the description of studies led in rodents it is highly ambiguous.

In the last part of this chapter the role of mature OLG in remyelination is put forward and most of the studies quoted here are from animal models and this could lead the reader to conclude that remyelination from mature OLG is a major process in rodents. In fact it is hardly visible in mice or at low levels and under specific conditions such as following training (Orthmann-Murphy et al., 2020; Bacmeister et al., 2020). Beside in the discussion about the properties of new myelin sheaths  formes from pre-existing OLG the work of Snaidero (202O) is mentioned but in this paper the authors only refer to newly formed OLG from reactive OPC.

So it should be more clearly stated that, with the available methods and data actually, it seems that remyelination in humans seems to occur preferentially thanks to pre-existing OLG where as in rodents the main repair process occurs through OPC reactivity and newly formed OLG.

Then the confusion is maintained when in chapter 9 the whole discussion to understand how to promote remyelination in MS is focused on OPC reactivity and differentiation (after stating in chapter 7 that remyelination in MS does not occur through OPC). This discrepancy would deserve discussion when the question of translation from animal models to MS therapy development is addressed.

 Why astrocytes do not appear in Figure 2? It would provide a more complete picture.
